# Assessing Measurement Properties of a Simplified Chinese Version of Sleep Condition Indicator (SCI-SC) in Community Residents

**DOI:** 10.3390/bs12110433

**Published:** 2022-11-03

**Authors:** Runtang Meng, Esther Yuet Ying Lau, Karen Spruyt, Christopher B. Miller, Lu Dong

**Affiliations:** 1School of Public Health, Hangzhou Normal University, Hangzhou 311121, China; 2Sleep Laboratory, Department of Psychology, The Education University of Hong Kong, Hong Kong 999077, China; 3Centre for Psychosocial Health, The Education University of Hong Kong, Hong Kong 999077, China; 4Centre for Religious and Spirituality Education, The Education University of Hong Kong, Hong Kong 999077, China; 5Université de Paris, NeuroDiderot, INSERM, Paris 75019, France; 6Sir Jules Thorn Sleep & Circadian Neuroscience Institute (SCNi), Nuffield Department of Clinical Neurosciences, University of Oxford, Oxford OX3 9DU, UK; 7Big Health Inc., 461 Bush St #200, San Francisco, CA 94108, USA; 8Department of Behavioral & Policy Sciences, RAND Corporation, Santa Monica, CA 90401, USA

**Keywords:** Sleep Condition Indicator, measurement invariance, psychometrics, community

## Abstract

Background: The study aimed to assess the measurement properties of a simplified Chinese version of the Sleep Condition Indicator (SCI-SC) in the community. Methods: A psychometric evaluation through an observational cross-sectional survey design was conducted. Community residents (N = 751) in Hangzhou, China completed the SCI-SC and the simplified Chinese version of the Sleep Quality Questionnaire (SQQ) in July 2021. Data were randomly split into a development sample (N = 375) for model development by exploratory factor analysis (EFA) and a holdout sample (N = 376) for validation by confirmatory factor analysis (CFA). Multi-group CFA (MGCFA) was used to assess configural, metric, scalar, and strict measurement invariance across gender, age, marital status, body mass index (BMI), napping habits, generic exercise, hobby, and administered survey. Moreover, statistical analyses were performed to determine the reliability (alpha and omega) and construct validity of the instrument. Results: Both factor analyses showed a stable solution with two dimensions of Sleep Pattern and Sleep-Related Impact. Good structural validity, robust internal consistency, and construct validity with the SQQ were demonstrated. There was evidence of strict invariance across gender, BMI, napping habits, generic exercise, hobby, and administered survey subgroups, but only metric and scalar invariances were established across age and marital status groups, respectively. Conclusions: The SCI-SC demonstrated promising psychometric properties, with high SQQ concordance and consistent structure of the original version. The SCI-SC can be used by sleep researchers as well as healthcare professionals in various contexts in detecting risks for insomnia disorder in the community.

## 1. Introduction

Good sleep is essential to good health, and conversely, insomnia affects health and quality of life [1,2]. Insomnia complaints are the most ubiquitous sleep–wake problems in both community-dwelling individuals and clinical populations, with approximately one-tenth of the population meeting the diagnostic criteria of clinical insomnia [3]. Associated with significant distress and impairment in daytime functioning [4,5], insomnia has a vast socioeconomic burden and impact on public health. However, assessment and treatment can be challenging in practice, as insomnia within each demographic group is commonly associated with multiple interacting psychiatric and medical comorbidities [2,6]. To arrive at an accurate evaluation of insomnia, healthcare providers preferably should assess the sleep complaints and patterns with a structured clinical interview, together with prospective sleep diary and actigraphy [3,7]. As an effective interview and objective measures demand expertise, time, and resources, there is an imminent need for proper assessment with reliable, valid, and brief screening instruments as a critical part of sleep or general health evaluation, for researchers as well as healthcare providers, and in particular general practitioners [8,9].

Based on the Diagnostic and Statistical Manual of Mental Disorders, Fifth Edition (DSM-5) and International Classification of Sleep Disorders, Third Edition (ICSD-3) diagnostic criteria, insomnia is marked by dissatisfaction with sleep quality and/or quantity, characterized by at least one of the following symptoms: difficulty initiating sleep, difficulty maintaining sleep, and early morning awakening. In addition, symptoms are to be present for at least three months despite adequate opportunity and circumstances for sleep, and with adverse consequences for daytime functioning. In view of such updates, the Sleep Condition Indicator (SCI) was newly developed using DSM-5 insomnia criteria [10,11,12]. The SCI is brief but versatile and yields a dimensional perspective on sleep quality, a visual profile of nighttime sleep disturbances and daytime symptoms, and indicative cut-off points for clinically significant insomnia [10]. With its updated nature, succinctness and yet comprehensive coverage, the measure is instrumental in identifying insomnia risks using the latest recognized insomnia criteria.

To date, the SCI has been translated and validated in at least seven languages: Romanian (2013) [13], Italian (2015) [14], traditional Chinese (2017) [9], French (2017) [8], Swedish (2019) [15], Persian (2019) [16], and Arabic (2021) [17]. According to COSMIN taxonomy of measurement properties [18], these reports have supported most measurement properties of the original English SCI (i.e., content, structural, construct [convergent and discriminant] validity; internal consistency, test–retest reliability) [10,19] using classical test theory and/or modern test theory such as Rasch model. Nevertheless, to our knowledge, there has not been a simplified Chinese version of the Sleep Condition Indicator (SCI-SC), which would be critical for cross-cultural comparisons on insomnia disorder involving Chinese Mainland. Specifically, due to slight differences between simplified Chinese and traditional Chinese, the traditional Chinese SCI is notably not applicable to Chinese Mainland people who speak Mandarin and write simplified Chinese. Moreover, the invariance of the existing seven versions of the SCI also remains to be examined.

To address these research gaps, the current study was designed to assess the measurement properties of the SCI-SC, newly adapted from the SCI-traditional Chinese version, to reveal whether or not it is a suitable instrument for insomnia risk-screening in community-based populations reading or speaking simplified Chinese. Most importantly, the measurement invariance of the SCI across different socio-demographic variables was assessed, to ascertain its applicability across different subgroups.

## 2. Materials and Methods

### 2.1. Participants and Procedures

Using the rule of thumb of 20 subjects per item for factor analysis of the eight-item SCI, at least 160 subjects were needed for each factor analysis [20]. Moreover, to conduct exploratory factor analysis (EFA), confirmatory factor analysis (CFA), and multi-group CFA (MGCFA), a sample size of at least 300 respondents was recommended [21]. Individuals who were able to read (self-administered survey) simplified Chinese or communicate (interviewer-administered) in Mandarin were recruited. Exclusion criteria: (1) those who had difficulty understanding the study procedures; (2) people who were taking medication for sleep disorders, had psychiatric diagnoses or substance abuse. We recruited 751 community residents who completed a paper-and-pencil survey in person during their health check-ups in a community health service center between 8 July 2021 and 21 July 2021.

### 2.2. Measures

#### 2.2.1. Sleep Condition Indicator (SCI)

The SCI is an eight-item scale comprising two subscales of Sleep Pattern (SP) with five items and Sleep-Related Impact [SRI, hereinafter called the ‘Daytime Impact (DI)’] with three items, respectively. Each item is assessed on a five-point Likert scale (0–4), that are summed up to a global score ranging between 0 and 32, with a higher score indicating better sleep and a lower risk of insomnia over the last month. Both the original (English) SCI and the traditional Chinese SCI, including its two-item short form (SCI_SF), have good psychometric properties [9,10]. Usage permission was obtained from the respective corresponding authors. This simplified Chinese SCI was adapted from the traditional Chinese SCI according to normative linguistic validation processes [9].

#### 2.2.2. Sleep Quality Questionnaire (SQQ)

The SQQ consists of ten items: using a five-point Likert scale (0–4) ranging from strongly agree to strongly disagree with six items assessing Daytime Sleepiness (DS) and four items evaluating Sleep Difficulty (SD) over the past month. Composite scores range from 0 to 40, with higher scores indicating poorer sleep quality. Both the original (Japanese and English) SQQ and the simplified Chinese SQQ have good psychometric properties [22,23,24,25].

### 2.3. Statistical Approach

#### 2.3.1. Structural Validity

The collected data (N = 751) was randomly split into a development and a holdout samples, half (N = 375) for model development, and the other half (N = 376) for validation. Parallel analysis (PA) is available to accurately determine the number of factors in EFA, then PA with promax rotation was conducted to explore a scale structure most appropriate in a development sample. The factorial validity of the model identified from EFA was assessed by CFA in a holdout sample. Fit indices of Comparative Fit Index (CFI), Tucker–Lewis index (TLI), standardized root mean square residual (SRMR), and root mean square error of approximation (RMSEA) and its corresponding 90% confidence interval (CI) were used. Specifically, CFI and TLI ≥ 0.90, and RMSEA and SRMR ≤ 0.08 indicated satisfactory structural validity [26,27]. Chi-square and p-value with greater sensitivity in sample size were also reported.

#### 2.3.2. Measurement Invariance

After confirming the two-factor structure of the SCI-SC, we tested measurement invariance (MI) on the total sample (N = 751) across the following grouping variables: gender (female vs. male), age (≤18 years vs. >18 years), marital status (married vs. non-married), body mass index (BMI) (thinness vs. normal vs. overweight vs. obesity), napping habits (yes vs. no), generic exercise (yes vs. no), and hobby (yes vs. no). MGCFA was used to examine four forms of increasingly restrictive hypotheses: configural, metric, scalar, and strict invariance. Configural, metric, scalar, and strict invariance correspondingly refer to the same factor structure, factor loadings, item intercepts, and item residuals across groups. Metric, scalar, and strict invariance were tested by sequentially placing constraints on the parameters (i.e., loadings, intercepts, and residuals) of the configural invariance model. We added cumulatively these constraints to each hypothesized MI model. To avoid an incomplete and potentially biased view of the tests of measurement invariance and their results, we compared nested models using the scaled differences in the log likelihood chi squared statistic (i.e., change in chi-square, Δ*χ*^2^) and absolute differences in the alternative fit indices (i.e., change in alternative fit indices, ΔAFIs). The Δ*χ*^2^ represents the standard approach; whereas, AFIs are modern alternatives that have the advantage of being independent of sample size. Despite the sample size of over 700, the models were also compared with a chi-square difference test being nonsignificant. More specifically, ΔCFI ≤ 0.010, ΔTLI ≤ 0.010, and ΔRMSEA ≤ 0.015 were considered of adequate invariance when the Δ*χ*^2^ is significant [28,29]. If at least two out of three change in fit indices meet the cut-off criteria and does not significantly worsen model fit, we considered that measurement invariance held [30]. Given that our SCI responses violated multivariate normality assumptions, we used maximum likelihood parameter estimates with standard errors and a mean-adjusted chi-square test statistic that are robust to non-normality (MLM) estimators [31,32].

#### 2.3.3. Short Form of the SCI (SCI_SF)

A stepwise linear regression model was conducted to identify a subset of items (independent variables) accounting for the greatest proportion of variance in the SCI total score (dependent variable), and to explore and assess the derivation of the SCI_SF as mentioned above [9,10].

#### 2.3.4. Construct Validity

Pearson correlation was performed between the two SCI domain scores and the global score with the SQQ to study the construct validity with the SQQ. We hypothesized that the SCI would be significantly strong negative correlation (> 0.50) with the SQQ, given that both instruments are supposed to measure similar constructs [18].

#### 2.3.5. Internal Consistency

Internal consistency was assessed by obtaining inter-item, item-subscale, item-total correlations, alpha (α) and omega (ω) values. Item-total correlations should be between 0.20 and 0.80 [33]. Alpha(s) and Omega(s) ≥ 0.70 [18,34].

### 2.4. Analysis Software

All data analyses were performed using R (v.4.1.2), RStudio (v.2021.09.1, Build 372), and JASP (v.0.16.1). Missing data was replaced by the mean or median of observed values because the number of missing responses (%) did not exceed 10% [35]. The multivariate normality tests, CFA and MGCFA, were conducted with package “*MVN* v.5.9” [36], “*lavaan* v.0.6-9” [37], and “*semTools* v.0.5-5” [38]. We ran the figure procedure described below with package “*ggplot2* v.3.3.5” [39].

## 3. Results

### 3.1. Sample Description

Seven hundred and fifty-one community residents were identified and approached. We recruited 680 adult community residents (90.85%) in Hangzhou, China, with married (54.99%), and the rest being single (42.74%), divorced (0.93%), and widowed (1.33%). In terms of BMI based on China standard [For adults (age > 18): BMI < 18.5, thinness; 18.5 ≤ BMI < 23.9, normal; 24 ≤ BMI < 27.9, overweight; BMI ≥ 28, obesity [40] and for school-age children and adolescents (6 ≤ age ≤ 18): the classification of BMI depending on gender and age and its details in the literature [41], participants fell into the categories of thinness (13.45%), normal (57.79%), overweight (21.57%), and obesity (7.19%). All samples are from the administered survey. Community residents completed the SCI-SC, the simplified Chinese SQQ, and the socio-demographic variables questionnaire in both interviewer (one fourth) and self-administered (three fourths) form, with speaking Mandarin and reading simplified Chinese, respectively. Additional participant characteristics are reported in Table 1. Missing data did not exceed 1.47% (11/751) among sample variables.

### 3.2. Descriptive Statistics

Mean, standard deviation (SD), skewness, and kurtosis for the SCI-SC items are presented in Table 2. Missing data did not exceed 0.266% (2/751) across the SCI-SC items. All items, subscales, short form, and total for the SCI-SC revealed normality with skewness of less than 2 and kurtosis less than 7 [42]. However, the data did not follow the multivariate normal distribution by Mardia’s test statistic (Mardia’s test = 1591.992 for Skewness and 25.839 for Kurtosis, all *p* < 0.001) [43]. Thus, we used MLM for non-normality in the subsequent analysis.

### 3.3. Structural Validity

EFA was used to identify scale structure on the SCI-SC. Using the development sample (N = 375), PA with promax rotation (Kaiser-Meyer-Olkin = 0.851; Bartlett’s test *χ*^2^ (28) = 1203.505, *p* < 0.001) yielded a two-factor solution, accounting for 52.7% of the total variance. Factor 1 (Sleep Pattern) comprised items 1, 2, 3, 4, and 8 with factor loading ranging from 0.470 to 0.778 and explained 25.7% of the variance. Factor 2 (Daytime Impact) comprised items 5, 6, and 7 with factor loading ranging from 0.743 to 0.865 and explained 27.0% of the variance. There were no cross-loadings on any item, hence loadings and cross-loadings were adequate (Table 3).

Using the holdout sample (N = 376), CFA was conducted on the two-factor structure identified by EFA. CFA fit indices were excellent (CFI = 0.959, TLI = 0.939, RMSEA = 0.069 [0.050, 0.088], and SRMR = 0.053, *χ*^2^ (19) = 52.678, *p* < 0.001). The EFA and CFA results supported the SCI factorial structure reported in the original design (i.e., five items on Sleep Pattern and three on Daytime Impact).

### 3.4. Measurement Invariance

Table 4 and Table 5 describe the results of the tests of measurement invariance hypotheses in the SCI-SC items across gender, age, marital status, BMI, napping habits, generic exercise, and hobby subgroups.

Gender: For gender group comparisons, the two-factor measurement model provided a good/acceptable model fit across gender subgroups when all the parameters were freely estimated across gender (male and female) subgroups, which implies that the assumption of configural measurement invariance was satisfied. Successive stepwise constraints of factor loadings, item intercepts, and item residuals to assess metric, scalar, and strict invariances revealed acceptable model fits with negligible changes in CFI, TLI, and RMSEA fit indices (ΔCFI < 0.010; approximately ΔTLI ≤ 0.010; ΔRMSEA < 0.015).

Age: Using the same process as described above, successively stricter constraints were tested to evaluate configural, metric, scalar, and strict measurement invariance across age (≤ 18 years and > 18 years) subgroups. The assumption of metric measurement invariance was satisfied with acceptable model fits as evidenced by negligible changes in model fit with more constraints on the model (∆CFI < 0.010; approximately ΔTLI ≤ 0.010; ∆RMSEA < 0.015). However, scalar and strict invariance were not supported by fit indices (scalar invariance: ∆CFI = −0.016 and ΔTLI = −0.013; strict: ∆CFI = −0.017) meeting requirement for good/acceptable fit.

Marital Status: To examine measurement invariance of the SCI-SC across marital status (married and non-married), the measurement models were fit to subgroups of married and non-married (singles, divorced and widowed) individuals. Configural, metric, scalar, and strict measurement invariance of the two-factor measurement model were supported with respect to good/acceptable fit indices across marital status. Successively stricter constraints on the factor loadings and item intercepts were tested to evaluate metric and scalar invariance, and it was shown that configural, metric, and scalar invariance could be assumed across marital status subgroups, as evidenced by negligible changes in fit indices for the stricter models (approximately ∆CFI ≤ 0.010; approximately ΔTLI ≤ 0.010; ∆RMSEA < 0.015). However, strict invariance did significantly worsen model fit (strict invariance: ∆CFI = −0.020).

BMI: Configural invariance across BMI subgroups (thinness, normal, overweight, and obesity) was supported as evidenced by good/excellent fit. Successively stricter constraints on the factor loadings, item intercepts, and item residual variances revealed that metric, scalar, and strict measurement invariance were supported by the data with negligible changes in fit indices across the four BMI subgroups (∆CFI < 0.010; approximately ΔTLI ≤ 0.010; ∆RMSEA < 0.015).

Napping Habits: Configural invariance across napping habits (yes and no) subgroups was supported by fit indices meeting requirement for good/acceptable fit. The assumptions of metric, scalar, and strict measurement invariance were satisfied with acceptable model fits as evidenced by negligible changes in model fit for the more constrained model (∆CFI < 0.010; approximately ΔTLI ≤ 0.010; ∆RMSEA < 0.015).

Generic Exercise and Hobby: Strict measurement invariance was established across generic exercise (yes and no) and hobby (yes and no) with good/excellent model fit indices as evidenced by negligible changes in model fit for the stricter models (∆CFI < 0.010; ΔTLI ≤ 0.010; ∆RMSEA < 0.015).

Administered Survey: Configural invariance across administered survey (self-administered vs. interviewer-administered) subgroups was supported by fit indices meeting requirement for good/acceptable fit. The assumptions of metric, scalar, and strict measurement invariance were satisfied with acceptable model fits as evidenced by negligible changes in model fit for the more constrained model (approximately ∆CFI < 0.010; ΔTLI ≤ 0.010; ∆RMSEA < 0.015).

Finally, this pattern of results suggests that comparisons of the SCI-SC across the two gender groups, two age groups, two marital status groups, four BMI groups, two napping habits groups, two generic exercise groups, two hobby groups, and two administered groups are valid.

### 3.5. Short Form of the SCI (SCI_SF)

Consistent with the original as well as the traditional Chinese versions, results from the regression model showed that a model with items 3 (standardized β = 0.542) and 7 (β = 0.495) best predicted the SCI total score (adjusted R^2^ change = 0.772 (0.574 ± 0.199), F_2,748_ = 1269.725, *p* < 0.001, with no serial correlation noted; Durbin-Watson statistic = 2.026). This SCI_SF was strongly correlated with the SCI total score (*r* = 0.872, *p* < 0.001) (Figure 1).

### 3.6. Construct Validity

The SCI-SC (*r* = −0.558, *p* < 0.001) and SCI_SF (*r* = −0.506, *p* < 0.001) were moderately correlated with the Chinese SQQ. Figure 1 showed correlations between the SCI scores and the SQQ scores, including correlations between its subscales scores, with generally weak to strong correlations in the range of 0.265 to 0.619 (*p* < 0.001).

### 3.7. Internal Consistency

The inter-item correlations ranged from *r* = 0.139 to 0.710, with an average inter-item correlation of *r* = 0.384. The item-total and item-subscale correlations ranged from 0.513 to 0.757 and 0.611 to 0.900, respectively (Figure 1). Internal consistency of the SCI-SC was good with an alpha of 0.817 (0.796, 0.836) and an omega of 0.799 (0.778, 0.821); that of the SCI and its short form was acceptable with an alpha of 0.587 (0.528, 0.640). Alpha and omega values on subscales of the SCI-SC ranged from 0.734 to 0.874, suggesting acceptable levels (Table 2).

## 4. Discussion

Our findings provided strong support for the psychometric performance of the SCI-SC in a heterogeneous community sample. To our knowledge, this is the first study to assess the psychometric qualities of the SCI in a sample reading and speaking simplified Chinese. The SCI-SC was found to perform well on tests of construct validity regarding sleep quality, of internal consistency, as well as of structural validity based on factor structure and measurement invariance across socio-demographically diverse groups.

The SCI factorial structure, same as the original and traditional Chinese version, was reproducible by community-based assessment. The EFA identified a two-factor model as hypothesized in the original version, without deletion of items, given the strong (>0.45) and unambiguous (i.e., cross-loadings only <0.20) loading. Item wording is unambiguous in referring to Sleep Pattern or Daytime Impact, contributing to item-factor loadings on the primary instead of any other factor. Such a clear factor structure with items 1, 2, 3, 4 and 8 loaded on Sleep Pattern as well as items 5, 6 and 7 loaded on Daytime Impact, is also noteworthily reproduced in the French and Persian versions [8,16]. Furthermore, the SCI-SC was found to have a factor structure similar to that of the original version, with the exception that item 7, “overall sleep disturbance,” loaded on both factors (Sleep Pattern and Daytime Impact) in the original SCI [10] but only loaded on the second factor (Daytime Impact) of the SCI-SC, which was in turn consistent with the finding of the traditional Chinese version [9], from which this SCI-SC version derived. On the contrary, a one-factor model using polychoric-based explorative factor analysis for parallel analysis has been reported in the Swedish version, supporting unidimensionality of the SCI [15]. To our knowledge, studies of the SCI in other languages (e.g., Romanian [13], Italian [14], and Arabic [17]) did not report on the factor structure using EFA. One potential reason behind such discrepancies might be that clinical investigations tend to focus more on reliability (internal consistency, temporal stability) and/or construct validity, but not structural validity. Based on the existing evidence, it could be tentatively concluded that the factor structure of the SCI was in general consistent across the different language versions available with minor differences in factor loadings and further cross-cultural validation.

We then tested the replicability of this two-factor model (identified by the EFA) using CFA, and the corresponding fit indices were robust. This finding is consistent with two studies on the Persian version, one using EFA and CFA [16] and another using only CFA testing the same Persian version [44]. In addition, the classical test theory applied in most studies is sample dependent which implies that psychometric properties may vary between studies [33]. The discrepancy on the EFA and CFA results among those studies could be explained by cultural differences between the samples, the language or the translation as different tests and validation samples were used. Further evaluation of the factor structure of the SCI is warranted.

In establishing the measurement invariance based on item- and subscale-level analyses, this study substantially advanced sleep research as it provided a validated tool to measure sleep condition based on the most updated diagnostic criteria of DSM-5, for a large population of varied socio-demographic groups reading simplified Chinese. Scalar invariance of the SCI-SC across gender, BMI, napping habits, generic exercise, and hobby as well as marital status subgroups was attained, excluding only age subgroups. Presumably, the lack of scalar invariance between adults and adolescents might be due to the notable differences in psycho-physical needs and the unequal sample sizes. Given that scalar invariance is often taken as a pre-requisite for valid comparisons of observed mean differences between groups [28,45], our findings of measurement invariance of the SCI-SC supported the use of its scores in group comparisons for future studies. Particularly strict invariance would be necessary when the research interest is about item uniqueness or reliability [45]. Although some argued that a lack of strict invariance may lead to biased comparisons [46], others counter-argued that strict invariance is often not considered as necessary for testing differences in factor structure or latent means [28]. Taken together, our pattern of findings supported sleep condition as a universal construct as measured by the SCI-SC.

In addition to factor structure and measurement invariance, the current study revealed other psychometric properties of the SCI-SC comparable to those reported in previous studies. More specifically, previous studies have supported implicit construct validity and association with both subjective and objective measures of sleep condition, good internal consistency, satisfactory test–retest reliability, and adequate discriminant validity based on the classical test theory [8,9,10,13,14,15,16,44], as well as no substantial differential item functioning, satisfactory mean square, sensitive response forms, promising separation, and reliability according to modern test theory (Rasch model) [44]. Similarly, the construct validity of the SCI-SC was demonstrated by its observed association with the Chinese SQQ and its subscales. The alpha and omega coefficients for the subscales and the total scale were generally high, indicating good internal consistency. Regarding the two-item SCI-SC (short form), although the small number of items would preclude high internal consistency [34], its brevity and high correlation with the full scale warrant its use as both a clinical and a research tool for a quick screening of insomnia risks across socio-demographic groups [19,47]. Moreover, incredibly low levels of missing values indicated clarity, readability, and acceptability of the SCI-SC.

Despite our efforts to conduct a rigorous psychometric assessment of the SCI-SC using a large community-based sample, our study encountered limitations worth noting. First, the sample was recruited by convenience sampling and from only one site, restricting the generalizability of the results. Our sample data suffered from imbalance in age distribution (minimum 12 to maximum 88 years) such that the average age is young adults. Notably, young adults may have different sleep patterns and insomnia criteria than elderly. This may limit the external validity of the SCI-SC to elderly samples. Ideally, data collection with random sampling and multiple sites would further strengthen the validity of the instrument. Future studies could also establish age- and sex-norms with more representative samples. Second, longitudinal measurement invariance and long-term stability (e.g., intraclass correlation coefficient) were not evaluated due to time and resource constraints. Third, we did not directly assess the measurement invariance between the simplified Chinese, the traditional Chinese, and the English versions of the SCI (i.e., cross-cultural comparisons) [48]. Nevertheless, the factor structure of the two Chinese versions and that of the original English version were essentially the same. Lastly, our study did not include a clinical sample with objective measures of sleep, or a structured interview to rule out other stress disorders or medical and psychological conditions. We are indeed collecting more data from a diverse population because the validation of a tool is an ongoing process [49].

## 5. Conclusions

The current study provides initial data supporting promising psychometric properties of the SCI-SC with adequate structural validity on factor structure and measurement invariance, good internal consistency and construct validity, as well as consistency of factor structure with the original version and the traditional Chinese version. Validated in a large community-based sample, the SCI-SC can be used routinely as an insomnia-specific screening measure to assist the identification of clinical and subclinical insomnia for further assessment of sleep disturbances and clinical monitoring. The validation of the two-item short form further enhanced its wide application in various clinical, educational, and community settings.

## Figures and Tables

**Figure 1 behavsci-12-00433-f001:**
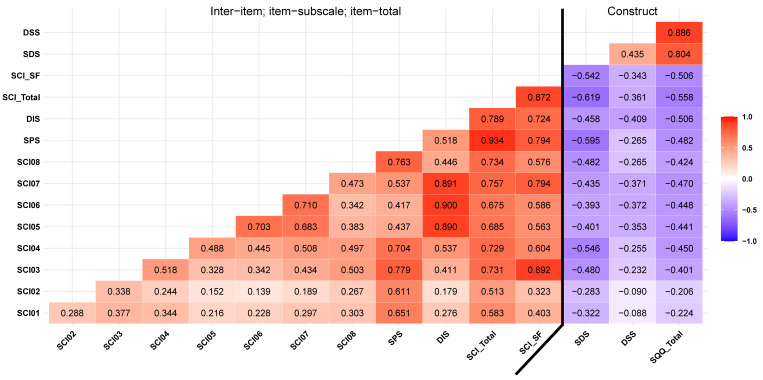
Inter−item, item−subscale, item−total, and constructs correlations between Sleep Condition Indicator and Sleep Quality Questionnaire. Note: Spearman correlations; SCI, Sleep Condition Indicator; SF, Short Form; SPS, Sleep Pattern Subscale; DIS, Daytime Impact Subscale; SCI01−08, Item 1−8; SQQ, Sleep Quality Questionnaire; DSS, Daytime Sleepiness Subscale; SDS, Sleep Difficulty Subscale.

**Table 1 behavsci-12-00433-t001:** Descriptive characteristics of study cohort.

Variables	N = 751	Missing (%)
Gender (n, % female)	426 (56.724)	1.198
Age (median, IQR)	28 (22.500)	0.266
Marital status (n, % married)	413 (54.993)	0.399
Body mass index (median, IQR)	22.039 (4.732)	0
Napping habits (n, % yes)	563 (74.967)	0.133
Generic exercise (n, % yes)	344 (45.806)	0
Hobby (n, % yes)	394 (52.463)	1.465
Administered (n, % Self-)	565 (75.223)	0

IQR, interquartile range.

**Table 2 behavsci-12-00433-t002:** Descriptive statistics for the SCI-SC (N = 751).

	Mean	SD	Skewness	Kurtosis	Alpha	Omega
01. How long does it take you to fall asleep?	2.996	1.164	−1.272	0.924	**0.813**	**0.798**
02. If you then wake up during the night, how long are you awake for in total?	3.298	1.190	−1.712	1.802	**0.826**	**0.812**
03. How many nights a week do you have a problem with your sleep?	3.237	1.203	−1.560	1.327	**0.785**	**0.771**
04. How would you rate your sleep quality?	2.470	0.886	−0.134	0.091	**0.785**	**0.758**
05. Poor sleep affected your mood, energy, or relationships?	2.968	0.901	−0.726	0.417	**0.791**	**0.791**
06. Poor sleep affected your concentration, productivity, or ability to stay awake?	2.956	0.904	−0.726	0.453	**0.793**	**0.794**
07. Poor sleep troubled you in general?	3.095	0.893	−0.941	0.803	**0.781**	**0.783**
08. How long have you had a problem with your sleep?	3.142	1.451	−1.443	0.429	**0.793**	**0.774**
Sleep Pattern	15.144	4.152	−1.079	0.595	0.734	0.746
Daytime Impact	9.019	2.413	−0.850	1.056	0.874	0.874
SCI_total	24.162	5.783	−0.923	0.559	0.817	0.799
SCI_SF	6.332	1.783	−1.224	0.933	0.587	N/A

SCI, Sleep Condition Indicator; SC, simplified Chinese; SF, short form; SD, standard deviation; N/A, not available; Dropped items in bold.

**Table 3 behavsci-12-00433-t003:** Exploratory factor analysis of the SCI-SC (N = 375).

	Factor 1, Sleep Pattern	Factor 2, Daytime Impact
SCI01	**0.563**	−0.016
SCI02	**0.470**	−0.050
SCI03	**0.778**	−0.039
SCI04	**0.625**	0.172
SCI05	−0.028	**0.865**
SCI06	−0.077	**0.863**
SCI07	0.155	**0.743**
SCI08	**0.646**	0.078
Variance	0.257	0.270

SCI, Sleep Condition Indicator; SC, simplified Chinese; SCI01-08, Item 1–8; Parallel analysis, Estimation method = Minimum residual; Applied rotation method = Promax; Factor loadings > 0.4 in bold.

**Table 4 behavsci-12-00433-t004:** Tests of measurement invariance of the SCI-SC across subgroups (N = 751).

Hypothesis	*χ^2^* (*df*)	*p*	Scaled Chi-Squared Difference Test		CFI	ΔCFI	TLI	ΔTLI	RMSEA (CI 90%)	ΔRMSEA
			Δ*χ^2^* (Δ*df*)	*p*						
Gender (female vs. male)										
Configural	105.380 (38)	<0.001			0.956		0.935		0.069 (0.055, 0.083)	
Metric	108.497 (44)	<0.001	2.905 (6)	0.821	0.958	0.002	0.946	**0.011**	0.062 (0.049, 0.076)	−0.007
Scalar	115.642 (50)	<0.001	5.258 (6)	0.511	0.957	−0.001	0.952	0.006	0.059 (0.046, 0.072)	−0.003
Strict	134.615 (58)	<0.001	18.896 (8)	0.015	0.950	−0.007	0.952	0.000	0.059 (0.048, 0.071)	0.000
Age (≤18 years vs. >18 years)										
Configural	93.658 (38)	<0.001			0.962		0.944		0.062 (0.048, 0.077)	
Metric	95.802 (44)	<0.001	4.742 (6)	0.577	0.965	0.003	0.955	**0.011**	0.056 (0.042, 0.070)	−0.006
Scalar	125.828 (50)	<0.001	50.235 (6)	<0.001	0.949	**−0.016**	0.942	**−0.013**	0.064 (0.051, 0.076)	0.008
Strict	158.709 (58)	<0.001	31.048 (8)	<0.001	0.932	**−0.017**	0.934	−0.008	0.068 (0.057, 0.080)	0.004
Marital Status (married vs. non-married)										
Configural	111.858 (38)	<0.001			0.950		0.927		0.072 (0.058, 0.086)	
Metric	114.214 (44)	<0.001	2.008 (6)	0.919	0.953	0.003	0.940	**0.013**	0.065 (0.052, 0.078)	−0.007
Scalar	135.564 (50)	<0.001	23.156 (6)	<0.001	0.943	−0.010	0.936	−0.004	0.068 (0.055, 0.080)	0.003
Strict	172.915 (58)	<0.001	34.781 (8)	<0.001	0.923	**−0.020**	0.926	−0.010	0.073 (0.062, 0.084)	0.005
BMI (thinness vs. normal vs. overweight vs. obesity)										
Configural	146.126 (76)	<0.001			0.954		0.933		0.070 (0.054, 0.086)	
Metric	162.495 (94)	<0.001	16.993 (18)	0.524	0.955	0.001	0.947	**0.014**	0.062 (0.047, 0.077)	−0.008
Scalar	187.336 (112)	<0.001	22.896 (18)	0.195	0.951	−0.004	0.951	0.004	0.060 (0.046, 0.074)	−0.002
Strict	201.283 (136)	<0.001	20.364 (24)	0.676	0.958	0.007	0.965	**0.014**	0.051 (0.037, 0.063)	−0.009
Threshold	N/A	>0.05	N/A	>0.05	>0.90	≤0.010	>0.90	≤0.010	<0.08	≤0.015

SCI, Sleep Condition Indicator; SC, simplified Chinese; df, degrees of freedom; CFI, comparative fit index; TLI, Tucker–Lewis index; RMSEA, root mean square error of approximation; CI, confidence interval; BMI, body mass index; Beyond the threshold in bold.

**Table 5 behavsci-12-00433-t005:** Tests of measurement invariance of the SCI-SC across subgroups (N = 751) (Continued).

Hypothesis	*χ^2^* (*df*)	*p*	Scaled Chi-Squared Difference Test		CFI	ΔCFI	TLI	ΔTLI	RMSEA (CI 90%)	ΔRMSEA
			Δ*χ^2^ *(Δ*df*)	*p*						
Napping Habits (yes vs. no)										
Configural	103.300 (38)	<0.001			0.957		0.936		0.068 (0.054, 0.082)	
Metric	105.323 (44)	<0.001	2.771 (6)	0.837	0.959	0.002	0.948	**0.012**	0.061 (0.048, 0.074)	−0.007
Scalar	109.004 (50)	<0.001	1.284 (6)	0.973	0.961	0.002	0.956	0.008	0.056 (0.043, 0.069)	−0.005
Strict	109.064 (58)	<0.001	6.639 (8)	0.576	0.966	0.005	0.967	**0.011**	0.048 (0.036, 0.060)	−0.008
Exercise (yes vs. no)										
Configural	89.550 (38)	<0.001			0.964		0.947		0.060 (0.046, 0.075)	
Metric	94.365 (44)	<0.001	4.491 (6)	0.611	0.965	0.001	0.956	0.009	0.055 (0.041, 0.069)	−0.005
Scalar	103.668 (50)	<0.001	8.471 (6)	0.206	0.963	−0.002	0.958	0.002	0.053 (0.040, 0.067)	−0.002
Strict	122.227 (58)	<0.001	18.268 (8)	0.019	0.955	−0.008	0.957	−0.001	0.054 (0.042, 0.066)	0.001
Hobby (yes vs. no)										
Configural	86.918 (38)	<0.001			0.967		0.952		0.059 (0.044, 0.073)	
Metric	96.350 (44)	<0.001	9.249 (6)	0.160	0.965	-0.002	0.956	0.004	0.056 (0.043, 0.070)	−0.003
Scalar	100.199 (50)	<0.001	1.845 (6)	0.933	0.967	0.002	0.963	0.007	0.052 (0.038, 0.065)	−0.004
Strict	107.410 (58)	<0.001	9.318 (8)	0.316	0.967	0.000	0.968	0.005	0.048 (0.035, 0.060)	−0.004
Administered Survey (self-administered vs. interviewer-administered)										
Configural	103.677 (38)	<0.001			0.952		0.930		0.068 (0.054, 0.082)	
Metric	112.127 (44)	<0.001	7.958 (6)	0.241	0.950	−0.002	0.937	0.007	0.064 (0.051, 0.077)	−0.004
Scalar	132.636 (50)	<0.001	22.555 (6)	<0.001	0.940	−0.010	0.933	−0.004	0.066 (0.054, 0.079)	0.002
Strict	160.884 (58)	<0.001	25.558 (8)	0.001	0.925	**−0.015**	0.928	−0.005	0.069 (0.058, 0.080)	0.003
Threshold	N/A	>0.05	N/A	>0.05	>0.90	≤0.010	>0.90	≤0.010	<0.08	≤0.015

SCI, Sleep Condition Indicator; SC, simplified Chinese; df, degrees of freedom; CFI, comparative fit index; TLI, Tucker–Lewis index; RMSEA, root mean square error of approximation; CI, confidence interval; BMI, body mass index; Beyond the threshold in bold.

## Data Availability

The datasets used and analyzed during the present study are available from the first author upon reasonable request. Anyone interested in using the formatted SCI-SC and its scoring rubric should also be directed to the first author.

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
