# Peer review of "Assessing Measurement Properties of a Simplified Chinese Version of Sleep Condition Indicator (SCI-SC) in Community Residents"

_behavsci, 2022, doi:10.3390/bs12110433_

Round 1
Reviewer 1 Report
Dear authors
Greetings
The findings presented for the SCI-SC show psychometric properties with high agreement of the SQQ, in addition to having a similar validity and structuring with the original version. In this way, I believe that studies for evaluation at a prospective level can be used to determine risk probabilities for insomnia disorders in communities.
Author Response
Thank you for your insightful views and constructive comments on our manuscript. We have made some revisions for a better presentation of our work. We are indeed collecting more data from a diverse population because the validation of a tool is an ongoing process. We equally agree that the SCI-SC could, with time, be used as a tool to determine insomnia risk.
Thank you for the time to read our work.

Reviewer 2 Report
Thank you for your manuscript. My comments and suggestions are as followings:
1. Younger adults have different sleep patterns and insomnia criteria from older adults. However, the average age of this study sample is younger adults, so does it apply to all adults?
2. Table 1 shows that approximately 25% of the sample is from the Administered Survey. However, regarding the statement, 'This is the first study to assess the psychometric qualities of the SCI in a sample reading and speaking simplified Chinese" in lines 329-330. Please clarify the consistency of the survey approach.
3. Line 77, "newly adapted from the SCI-traditional Chinese version," please clarify whether the pre-test was conducted on the renewed simplified Chinese version? If so, how did this study conduct it?
4. Please specify the method of reliability and validity from the traditional Chinese version of SCI while converting SCI into the simplified Chinese version.
5. Table 4 shows "Age (≤ 18 years vs. > 18 years)" did the analysis exclude underage people? Again, please verify the definition of this variable.

Author Response
Review Report (Reviewer 2)
Thank you for your manuscript. My comments and suggestions are as followings:
Thank you for your pertinent comments, constructive feedback, and practical suggestions. We revised our manuscript with tracked changes based. Below please find our point-by-point responses to the comments. We hope to have addressed your comments satisfactorily. Please let us know if we need to provide anything else regarding this revision. We look forward to your favorable decision.
1) Younger adults have different sleep patterns and insomnia criteria from older adults. However, the average age of this study sample is younger adults, so does it apply to all adults?
We apologize for the confusion generated by the aapplied wording and sincerely hope that our logic is now easier to follow with new wording. The distribution of our sample data suffered from imbalance in age group size, hence, it was added to the limitations (Page 12).
2) Table 1 shows that approximately 25% of the sample is from the Administered Survey. However, regarding the statement, 'This is the first study to assess the psychometric qualities of the SCI in a sample reading and speaking simplified Chinese" in lines 329-330. Please clarify the consistency of the survey approach.
Thank you for capturing this aspect. All samples (N = 751) are from the Administered Survey. Community residents completed the SCI-SC, the simplified Chinese SQQ and the socio-demographic variables questionnaire in both interviewer- (25%) and self-administered (75%) forms, with speaking Mandarin and reading simplified Chinese, respectively. We have modified this on lines 193-197, Page 4-5.
3) Line 77, "newly adapted from the SCI-traditional Chinese version," please clarify whether the pre-test was conducted on the renewed simplified Chinese version? If so, how did this study conduct it?
Thank you for this comment. Before collecting data on community residents in 2021, the SCI-SC as a criterion, we conducted a large-scale assessment project among university students, healthcare workers, and general patients (total sample, N = 13,325) in another study (Refs: A and B, see below) from 2019 to 2020. There were no misleading claims such as false, unclear, unintelligible, or ambiguous information on the SCI-SC in the large-scale survey. This large-scale survey may treat as the pre-test that was conducted on the renewed simplified Chinese SCI (SCI-SC). More details see
Refs:
- A) Meng, R., Kato, T., Mastrotheodoros, S., Dong, L., Fong, D. Y. T., Wang, F., ... & Gozal, D. (2022). Adaptation and validation of the Chinese version of the Sleep Quality Questionnaire. Quality of Life Research, 1-14. https://doi.org/10.1007/s11136-022-03241-9
- B) Meng, R. (2020). Development and Evaluation of the Chinese Version of the Sleep Quality Questionnaire (doctoral dissertation in Chinese). Wuhan University, Wuhan.
4) Please specify the method of reliability and validity from the traditional Chinese version of SCI while converting SCI into the simplified Chinese version.
Thanks for this comment. Both the original (English) SCI and the traditional Chinese SCI, including its 2-item short form (SCI_SF), have good psychometric properties (i.e. reliability and validity). In the manuscript we refer to Refs 9 and 10.
Refs:
9) Wong, M. L., Lau, K. N. T., Espie, C. A., Luik, A. I., Kyle, S. D., & Lau, E. Y. Y. (2017). Psychometric properties of the Sleep Condition Indicator and Insomnia Severity Index in the evaluation of insomnia disorder. Sleep medicine, 33, 76-81. https://doi.org/10.1016/j.sleep.2016.05.019
10) Espie, C. A., Kyle, S. D., Hames, P., Gardani, M., Fleming, L., & Cape, J. (2014). The Sleep Condition Indicator: a clinical screening tool to evaluate insomnia disorder. BMJ open, 4(3), e004183. http://dx.doi.org/10.1136/bmjopen-2013-004183
5) Table 4 shows "Age (≤ 18 years vs. > 18 years)" did the analysis exclude underage people? Again, please verify the definition of this variable.
Thank you for the comment. The age range goes from minimum 12 years to maximum 88 years (the descriptive Statistics are : median = 28, IQR = 22.5; mean = 33.879, SD = 16.544). We have split thesample into those with Age ≤ 18 years being N = 71 and those with age > 18 years being N = 680)”. Therefore, we have not excluded underaged people.

Reviewer 3 Report
This study is a validation study of the SCI-SC scale in the community. And the authors assessed the measurement invariance of the SCI-SC across different socio-demographic variables. This study provides important evidence for clinical and epidemiological studies, but minor points need to be corrected.
An explanation of why SCI-SC is needed should be added in the background; the difference between SCI-SC and the SCI-Traditional Chinese version is difficult to understand for non-Chinese speaking readers.
L173
A numerical value should be added for the classification of BMI: Thinness: BMI < 18.5?
L237
“However, scalar and strict invariance were not supported by fit indices meeting requirement for good/acceptable fit.”
Which fit indices in the table are indicated?
Similarly, fit indices for other characteristics variables should be clarified in the text.
L319
Is 0.265 not 0.088? Please confirm.
Author Response
Review Report (Reviewer 3)
This study is a validation study of the SCI-SC scale in the community. And the authors assessed the measurement invariance of the SCI-SC across different socio-demographic variables. This study provides important evidence for clinical and epidemiological studies, but minor points need to be corrected.
Thank for your time and insightful comments. We hope to have addressed your comments satisfactorily.
1) An explanation of why SCI-SC is needed should be added in the background; the difference between SCI-SC and the SCI-Traditional Chinese version is difficult to understand for non-Chinese speaking readers.
Thank you for this comment. We have modified this sentence in the manuscript as follows(Line 77-80, Page 2: “However, due to slight differences between simplified Chinese (Chinese Mainland) and traditional Chinese (Hong Kong, Macao and Taiwan, China), the traditional Chinese SCI is notably not applicable to Chinese Mainland people who speak Mandarin and write simplified Chinese” In Chinese Mainland, Mandarin is the spoken language and people use Simplified Chinese when they write. In Hong Kong and Macao, Cantonese is the predominant dialect while people write in Traditional Chinese. The exception is Taiwan, where people speak Mandarin and write in Traditional Chinese.
2) L173
A numerical value should be added for the classification of BMI: Thinness: BMI < 18.5?
Thank you for capturing this oversight. We indeed used BMI based on China standard. “Thinness: BMI < 18.5” was included in adults for the classification of BMI.
For adults (age > 18): BMI < 18.5, thinness; 18.5 £ BMI < 23.9, normal; 24 £ BMI < 27.9, overweight; BMI ³ 28, obesity and for school-age children and adolescents (6 £ age £ 18): the classification of BMI depending on gender and age and its details in the literature (Line 189-192, Page 4).
Refs:
â‘ Ministry of Health of the People's Republic of China, Guidelines for the Prevention and Control of Overweight and Obesity in Chinese Adults (in Chinese). Acta Nutrimenta Sinica 2004, 26, (1), 1-4.
â‘¡ National Health Commission of the People's Republic of China, Screening for overweight and obesity among school-age children and adolescents (in Chinese). In Health Industry Standard of the People's Republic of China (WS/T 586—2018), National Health Commission of the People's Republic of China: Beijing, 2018.
3) L237
“However, scalar and strict invariance were not supported by fit indices meeting requirement for good/acceptable fit.”
Which fit indices in the table are indicated?
Similarly, fit indices for other characteristics variables should be clarified in the text.
Thank you for this comment. We clarified fit indices in the text (Page 6 and Page 7). Additionally, the fit indices that exceed the critical values are put in bold in Table 4.
4) L319
Is 0.265 not 0.088? Please confirm.
Thank you for capturing this.
In point 3.6. presented is the structural validity. Following the COnsensus-based Standards for the selection of health Measurement INstruments (COSMIN) taxonomy and guideline (Refs.: â‘ and â‘¡, see below), construct validity hypotheses on correlations between domains (i.e. involving scales or subscales, instead of items). The correlation value of 0.088 is between item (SCI01) and subscale (DSS). Hence, we confirm that 0.088 is the correct value..
Refs.:
â‘ Prinsen, C. A., Mokkink, L. B., Bouter, L. M., Alonso, J., Patrick, D. L., De Vet, H. C., & Terwee, C. B. (2018). COSMIN guideline for systematic reviews of patient-reported outcome measures. Quality of life research, 27(5), 1147-1157. https://doi.org/10.1007/s11136-018-1798-3
â‘¡ Mokkink, L. B., Terwee, C. B., Patrick, D. L., Alonso, J., Stratford, P. W., Knol, D. L., ... & de Vet, H. C. (2010). The COSMIN study reached international consensus on taxonomy, terminology, and definitions of measurement properties for health-related patient-reported outcomes. Journal of clinical epidemiology, 63(7), 737-745. https://doi.org/10.1016/j.jclinepi.2010.02.006.
